# Bioactive Compounds from Cardoon as Health Promoters in Metabolic Disorders

**DOI:** 10.3390/foods11030336

**Published:** 2022-01-25

**Authors:** Luís R. Silva, Telma A. Jacinto, Paula Coutinho

**Affiliations:** 1CPIRN-UDI/IPG—Centro de Potencial e Inovação em Recursos Naturais, Unidade de Investigação para o Desenvolvimento do Interior do Instituto Politécnico da Guarda, 6300-559 Guarda, Portugal; telmajacinto@ipg.pt (T.A.J.); coutinho@ipg.pt (P.C.); 2CICS-UBI—Health Sciences Research Centre, University of Beira Interior, 6201-506 Covilhã, Portugal

**Keywords:** *Cynara cardunculus* L., phytochemical composition, metabolic disorders, Mediterranean plant, biological activities

## Abstract

Cardoon (*Cynara cardunculus* L.) is a Mediterranean plant and member of the Asteraceae family that includes three botanical taxa, the wild perennial cardoon (*C. cardunculus* L. var. *sylvestris (Lamk) Fiori*), globe artichoke (*C. cardunculus* L. var. *scolymus L. Fiori*), and domesticated cardoon (*C. cardunculus* L. var. *altilis DC.*). Cardoon has been widely used in the Mediterranean diet and folk medicine since ancient times. Today, cardoon is recognized as a plant with great industrial potential and is considered as a functional food, with important nutritional value, being an interesting source of bioactive compounds, such as phenolics, minerals, inulin, fiber, and sesquiterpene lactones. These bioactive compounds have been vastly described in the literature, exhibiting a wide range of beneficial effects, such as antimicrobial, anti-inflammatory, anticancer, antioxidant, lipid-lowering, cytotoxic, antidiabetic, antihemorrhoidal, cardiotonic, and choleretic activity. In this review, an overview of the cardoon nutritional and phytochemical composition, as well as its biological potential, is provided, highlighting the main therapeutic effects of the different parts of the cardoon plant on metabolic disorders, specifically associated with hepatoprotective, hypolipidemic, and antidiabetic activity.

## 1. Introduction

Mediterranean plants have been used since ancient times by humans; the knowledge about these plants and their benefits has been accumulated and transmitted over several generations. Mediterranean plants are used for many purposes, such as medicine, food, spices, and ritual components [1,2]. In addition to their nutritional value, Mediterranean plant-based foods are seen as a simple way to introduce the concepts of wellness and health in our daily diet via the consumption of natural products. This is associated with the content of various bioactive compounds, such as phenolic compounds, inulin, vitamins, and minerals, being described as possessing great biological potential, such as antioxidant, anti-inflammatory, antimutagenic, anticancer, and neuroprotective activity [3,4,5,6].

One of the most prominent Mediterranean plants in recent years has been *Cynara cardunculus* L., due to its high biological and industrial potential [7,8,9]. 

Cardoon has been widely used in the Mediterranean diet and folk medicine since ancient times, being described as an important source of several bioactive compounds, such as fiber, inulin, minerals, phenolic compounds, and sesquiterpene lactones [10,11,12]. The qualitative composition of phenolics can be mainly associated with the hydroxycinnamic acid and flavonoid classes. With respect to hydroxycinnamic acids, 5-*O*-caffeoylquinic acid (chlorogenic acid), 1,5-*O*-dicaffeoylquinic acid, and 3,5-O-dicaffeoylquinic acid are reported as the most abundant. Additionally, apigenin and luteolin (both present as glucoside andrutinosides) are the main flavonoids present in cardoon [7,8,13]. Lastly, cyanidin caffeoylglucoside derivatives have also been reported in some cultivars [7,8]. Given this composition, cardoon has important nutritional value and therapeutic properties, such as antioxidant, antibacterial, antifungal, anti-HIV, lipid-lowering, cytotoxic, anti-inflammatory, antidiabetic, antihemorrhoidal, cardiotonic, and choleretic activity [6,7,8,14,15,16].

This comprehensive review provides an overview of the cardoon nutritional and phytochemical composition, as well as its biological potential. Furthermore, it also discusses the most recent literature regarding the main therapeutic effects of different parts of the cardoon plant on metabolic disorders. 

## 2. *Cynara cardunculus* L.

*Cynara cardunculus* L. belongs to the Asteraceae family, one of the largest families of the plant kingdom. *Cynara* is a relatively small genus, native to the Mediterranean basin [11]. *Cynara cardunculus* comprises three taxa: (i) the wild cardoon (*Cynara cardunculus* L. var. *sylvestris*), (ii) the domesticated cardoon (*Cynara cardunculus* var. *altilis* DC), and (iii) globe artichoke (*Cynara cardunculus* var. *scolymus* L.) [9,15,17]. Studies on the morphology and phytogeography of the *Cynara* genus reported that the referred plants belong to a single species and should consequently be classified as subspecies [11]. With respect to cultivated cardoon, its production is more limited to southern Europe, namely, France, Greece, Italy, Portugal, and Spain. Additionally, *C. cardunculus* L. is distributed across the Mediterranean basin, Macaronesia (Madeira and Canary Islands), North Africa, Cyprus, and Turkey, and it has also colonized Mexico, California, Argentina, Chile, Peru, Australia, China, and West Africa [11]. In these areas, cardoon has revealed its high adaptability to the climate conditions, i.e., a semiarid bioclimate, allowing its cultivation as a perennial crop in marginal lands with low moisture conditions (drylands).

Cardoon is a perennial plant that can reach more than 2 m in height with a thick and rigid stem, often branched in the upper parts; it is longitudinally ribbed and covered in cotton down. This plant has an annual development cycle, which can start with seed germination in autumn or spring, being completed in summer. The above-plant portion of the plant dies down each year, but its large taproot regenerates each year, growing up to 1 m down [10,18].

The seeds are 6–8 mm long, four-sided, and smooth. Their color is light gray, brown, or black, sometimes with longitudinal streaks. A circle of feathery hairs is present at the top of each seed, 25–40 mm in length, that readily falls off. 

The leaves form a strong and large basal rosette and can be up to 120 by 30 cm. The upper leaves are comparatively smaller, 10–50 cm in length. Furthermore, they are grayish-green on the upper surface and slightly hairy. On the underside, they are covered in dense hair giving a white woolly appearance. Each leaf is deeply lobed, with each lobe often partly divided again. The tips of the leaves are spiked with yellowish/orange spines, 5–20 mm long [10,18,19].

The inflorescence can be called a capitula or head, and there are hermaphrodites and tubular flowers (florets). The inflorescence occurs singly at the top of a branch on a thick stalk, 1–6 cm long. The flower heads are almost round in shape and grow to be 4–5 cm across. They consist of blue, pink, or purple florets arranged on a fleshy receptacle, enclosed by several large bracts. The bracts are purplish in color and taper to the end in a flattened spine [10,18,19].

Globe artichoke is cultivated for its edible phyllary bases and receptacles of its floral heads. Leaf stalks and leaf bases can be eaten as a cooked vegetable (boiled, braised, or sautéed) and used in salads and soups; they have an artichoke-like flavor. Additionally, artichoke flower extract is used in the food industry as an additive to prevent oil autooxidation and rancidity [20].

On the other hand, cardoon stalks are eaten and their flowers are used as plant rennet for the manufacture of French, Italian, Spanish, and Portuguese cheeses; this is legally required for the production of protected designation of origin (PDO) cheeses from Spain and Portugal, due to their richness in different proteases [10,11,21,22,23]. Additionally, cardoon is utilized by the paper industry due to its high cellulose and hemicellulose contents. It is also applied for biomass and bioenergy production (biogases as bioethanol and biomethane), as it includes 2–3 t·ha^−1^ of seeds, which are a good source of proteins and edible oil [7,8,24,25,26], as well as a good source of oil for biodiesel production [27]. The seeds contain around 25% oil, with good alimentary quality. The seed pomace after the extraction can be used for animal feed [10].

Additionally, cardoon leaf extract proved to have a great inhibitory effect against seed germination from weeds, revealing its potential to be used as a bioherbicide [28]. In a similar line, Restuccia and colleagues [29] combined a biocontrol yeast *W. anomalus* BS91 and three extracts (aqueous, methanolic, and ethanolic) from *C. cardunculus* L. var. *altilis* DC. leaves, the results revealed stronger protection with a combination of the ethanolic extract against green mold decay on oranges and lemons than when used singularly. These results indicate that biocontrol agents and leaf extracts, used in combination, can provide a stronger protection than when used singularly.

Moreover, the cytotoxicity of cardoon has been studied by several authors, reporting that cardoon does not present any toxicity toward noncancer cells; on the contrary, the cytotoxicity against cancer cells can depend on the plant part, maturity stage, and genetic background [30,31]. 

Cardoon produces a considerable amount of field residues; 590 days after planting, biomass accumulation may reach 3968 g·plant^−1^ [32]. Stalks, capitula, and leaves represent an average of 40%, 35%, and 25% of the aboveground biomass, respectively, and they possess a low moisture content (10–15%) [19,33], with a yield per hectare ranging from 3.64–38.38 t/year, with an average of 19.22 t/year during the first 3 years of production [34,35,36], supporting a good yield as a source of bioactive compounds [10,13,14]. 

## 3. Nutritional and Chemical Composition

The edible portion of the cardoon plant is the immature inflorescence (capitula or heads), including the inner bracts and the upper part of the receptacle, which can be eaten as fresh, canned, or frozen vegetal after minimal processing [15,37,38]. Cardoon is extremely nutritive, containing carbohydrates (inulin, sugars, and fiber), proteins and amino acids, fatty acids, organic acids, minerals, and vitamins. It is also an important source of bioactive compounds with pharmaceutical potential, including phenolic compounds, phytosterol, and volatiles [8,9,10,15,32,37,38,39]. The recognized high nutritive value of cardoon heads is attributed to the low content of lipids and high content of minerals, vitamins, and bioactive compounds, which are potent natural antioxidants substances [10,15]. The roots and rhizomes provide a source of inulin [32,39,40], an enhancer of the human intestinal flora, and the leaves provide a source of antioxidants, mainly dicaffeoylquinic acids and luteolin [9,15]. Furthermore, cardoon leaves are traditionally used in popular medicine as diuretics, hepatoprotectors, cholagogues, choleretics, and antidiabetics [41].

The chemical composition has been studied by various researchers in different biomass fractions: root [32,39]; stalks [42]; capitula fractions (receptacle, bracts, hairs, and pappi) [43]; heads [7]; leaves [35,44]; flowers [45]; seeds [7,46,47], as summarized in Table 1, Table 2, Table 3, Table 4, Table 5, Table 6, Table 7, Table 8 and Table 9.

According to the USDA [48], cardoon is mostly composed of water (94% for raw cardoon) and possesses a low caloric content with 17 kcal/100 g. With respect to the dry matter, the literature reports that the content varies significantly with the plant biomass material. The contents of raw cardoon are around 1.13% total ash content, 4.07% total sugars, 1.6% total dietary fiber, 0.7% total proteins, and 0.1% total lipids (Table 1).

### 3.1. Macronutrients

The macronutrients, carbohydrates, proteins, and fatty acids, have a unique set of properties that influence health, but all provide energy and essential components to sustain life; their combination in our diet is fundamental to maintain longevity and health [49]. The World Health Organization (WHO) reports a dietary recommendation of percentage energy contribution to a diet for the prevention of chronic diseases, including carbohydrate (50–75%), fat (15–30%), and protein (10–15%) [50]. Cardoon is rich in carbohydrates, proteins, and fatty and organic acids (Table 1, Table 2, Table 3, Table 4, Table 5 and Table 6).

#### 3.1.1. Carbohydrates

Carbohydrates are the most common biomolecules in the world, defined as simple sugars.

The free sugars found in cardoon’s vegetal parts are reported in Table 2. In general, sucrose was the sugar reported in higher concentrations (seeds, bracts, and heads), with contents ranging from “not detected” to 8.77 g/100 g of d.w. [7,43,46]. Sugars play an important role in the defense reactions of plants. Additionally, it has been reported that glucose, sucrose, and trehalose can regulate an important number of processes related to plant metabolism and growth, acting independently of the basal functions; furthermore, they can also act as signaling molecules [51].

Another important sugar present in higher amounts in the roots of cardoon is inulin, with contents ranging from 22.4 to 49.6 g/100 g d.w. [32,39] (Table 2), being considered the main reserve sugar in roots of *C. cardunculus* L. [32]. Inulin, a linear fructan, i.e., a polymer of fructose units, usually with a glucose molecule at the end, is a reserve carbohydrate that is accumulated mainly in the roots and tubers of several plants of the Asteraceae family, such as cardoon [52]. This polymer has been exploited by nutraceutical and pharmaceutical industries and for other biorefining applications [32,39,40]. Several purposes have been reported, such as prebiotics, a nutritional supplement as low caloric soluble dietary fiber, and a mediate sugar for lipid metabolism in diabetic and hypercholesterolemia [53]. Additionally, inulin can be used as a diagnostic agent for the determination of kidney function [54], for the production of fructose by enzymatic hydrolysis of inulin from inulin-rich biomass, and for ethanol production [55,56,57].

**Table 1 foods-11-00336-t001:** Nutritional composition of raw cardoon [48].

Nutrient (Unit)	Raw
Basic chemical composition	
Water (g/100 g)	94
Energy (kcal/100 g)	17
Energy (kJ/100 g)	71
Macronutrients	
Total protein (g/100 g)	0.7
Total lipids (g/100 g)	0.1
Fatty acids, total saturated (g/100 g)	0.011
SFA 16:0 (g/100 g)	0.009
SFA 18:0 (g/100 g)	0.002
Fatty acids, total monounsaturated (g/100 g)	0.018
MUFA 16:1 (g/100 g)	0
MUFA 18:1 (g/100 g)	0.018
MUFA 20:1 (g/100 g)	0
Fatty acids, total polyunsaturated (g/100 g)	0.041
PUFA 18:2 (g/100 g)	0.041
Carbohydrates (g/100 g) (by difference)	4.07
Total ash (g/100 g)	1.13
Total dietary fiber (g/100 g)	1.6
Total sugars (g/100 g)	4.07
Micronutrients	
Minerals	
Calcium, Ca (mg/100 g)	70
Iron, Fe (mg/100 g)	0.7
Magnesium, Mg (mg/100 g)	42
Phosphorus, P (mg/100 g)	23
Potassium, K (mg/100 g)	400
Sodium, Na (mg/100 g)	170
Zinc, Zn (mg/100 g)	0.17
Cooper, Cu (mg/100 g)	0.23
Manganese, Mn (mg/100 g)	0.256
Fluoride, F (µg/100 g)	0.2
Vitamins	
Vitamin C (mg/100 g)	2
Thiamin (mg/100 g)	0.02
Riboflavin (mg/100 g)	0.03
Niacin (mg/100 g)	0.3
Pantothenic acid (mg/100 g)	0.338
Vitamin B6 (mg/100 g)	0.116
Folate, total (µg/100 g)	68
Folate, DFE (µg/100 g)	68
Folate, food (µg/100 g)	68
Vitamin A, RAE (µg/100 g)	0
Vitamin A, IU (IU/100 g)	0
Vitamin B12 (µg/100 g)	0
Vitamin D (D2 + D3) IU (IU/100 g)	0
Vitamin D (D2 + D3) (µg/100 g)	0

**Table 2 foods-11-00336-t002:** Free sugar composition of cardoon vegetal parts.

Free Sugars (g/100 g d.w.)	Raw ^A^	Roots ^B^	Flowers ^C^	Seeds ^D^	Bracts ^E^	Heads ^F^
Fructose	-	0.1–6.47	-	n.d.–1.94	0.14–1.41	0.013–0.51
Glucose	-	0.07–4.54	2.47–4.69	n.d.–0.78	0.10–0.557	0.0–2.02
Sucrose	-	0.16–4.39	-	0.30–8.77	0.12–4.970	n.d.–2.39
Trehalose	-	-	-	0.16–36.44	0.24–1.16	0.23–0.98
Raffinose	-	-	-	n.d-1.31	n.d.–2.13	0.0–2.62
Rhamnose	0.75–1.1	-	1.10–1.19	-	-	-
Arabinose	1.17–2.7	-	4.23–6.03	-	-	-
Xylose	21.49–27.0	-	2.12–2.61	-	-	-
Mannose	1.1–1.8	-	0.44–0.52	-	-	-
Galactose	1.35–2.5	-	1.27–1.82	-	-	-
Inulin	-	22.4–49.6 *	-	-	-	-

n.d., not detected; * values for roots with 2 years; -, there are no data. ^A^ [58,59]; ^B^ [32,39]; ^C^ [45]; ^D^ [7,46,47]; ^E^ [43]; ^F^ [7].

Lignocellulosic biomass (cellulose, hemicellulose, and lignin) is the most abundant renewable resource on earth. This biomass is resistant to enzymatic and chemical degradation and possesses a protective function against natural agents and microbial attacks [59,60]. Cardoon is a lignocellulosic plant, in which lignocellulosic residues represent around 70–80% of the cardoon plant [61]. The contents of their vegetal parts range from 4.6 to 55.2 g/100 g d.w. Stems were revealed to have higher amounts (Table 3) [10,35,45,58,60,62,63].

Lignocellulose is naturally recalcitrant, whereby enzymatic hydrolysis of cellulose is extremely difficult compared to the breakdown of other renewable vegetal materials, such as starch [58,60]. Developing methods for primary fractionation depends on their economic assessment and environmental impact [58,59]. 

**Table 3 foods-11-00336-t003:** Fiber composition of cardoon vegetal parts.

Fiber (g/100 g d.w.)	Raw ^A^	Stems ^B^	Stalks ^C^	Leaves ^D^	Flowers ^E^
Hemicellulose	12.8–18.19	18.7–19.1	47.3–55.2	4.6–11.3	16.30–17.16
Cellulose	30.52–41.9	50.4–51.7	17.9–27.0	29.0–34.3	10.33–17.73
Lignin	14.21–18.9	10.7–11.9	13.3–28.8	8.7–12.5	7.60–10.73

^A^ [58,60]; ^B^ [35]; ^C^ [10,59,62,63]; ^D^ [35]; ^E^ [45].

Cellulose is probably the most abundant and valuable component of lignocellulosic biomass; it is a long-chain polymer, described as the most structural component in plant cells and tissues [64,65]. Its content in cardoon vegetal parts varies between 10.33% and 51.7%, with its greatest content being found in the stems [10,35,45,58,60,62,63]. Cellulose is a versatile polymer, being utilized by several sectors, such as veterinary foods, wood and paper, fiber and clothes, and cosmetics and pharmaceuticals as an excipient [65]. Cellulose and cellulose derivatives play important roles in pharmaceuticals, with properties related to extended and delayed-release coated dosage forms, extended and controlled release matrices, bioadhesives and mucoadhesives, compression tablets as compressibility enhancers, osmotic drug delivery systems, liquid dosage forms as thickening agents and stabilizers, granules and tablets as binders, semisolid preparations as gelling agents, and several other applications, as reviewed by Shokri and Adibkia [65].

The hemicellulose of cardoon is composed of 90–95% xylan polysaccharide, which may be used as an important source of xylose, a raw material for xylitol production [59]. Moreover, the residual cellulose is a great source of cellulosic fibers, which can be submitted to a saccharification process to obtain a valuable source of fermentable sugars, such as glucose, used for bioethanol production and other industrial applications [59]. The content of hemicellulose in cardoon materials varies between 4.6% and 55.2% (Table 3), with the stalks having the greatest content [10,35,45,58,60,62,63].

Lignin is another important chemical compound that requires proper and adequate valorization. Cardoon vegetal parts possess amounts of lignin that range from 8.7–28.8%, with the stalks being richest in lignin [10,35,45,58,60,62,63]. This valorization may be aimed at producing biobased products, such as antioxidants and polymers, while their use as resins is a very attractive market. Furthermore, lignin, mainly as lignosulfates, is used as a binding dispersant agent, with application in different industries [33,66].

Crop residues and forest products, as a rentable raw material in view of the principles of the circular economy and their economic impact, are very important, mainly for biorefinery approaches with an integral valorization of the biomass for bioethanol production and other high-value products [67].

#### 3.1.2. Proteins and Amino Acids

Proteins are chains of amino acids linked via amide bonds (commonly -known as peptide linkages). Proteins have many roles in the human body, like helping to repair and build our body’s tissues, providing a structural framework, and maintaining the pH and fluid balance. Furthermore, they also keep our immune system strong, transport and store nutrients, and can act as a source of energy, if needed.

Amino acids are defined as organic compounds containing both amino and acid groups. They are classified as essential or nonessential. Essential amino acids must be supplied from our exogenous daily diet because the human body lacks the metabolic pathways necessary to synthesize them [68,69]. 

Cardoon presents a low content of proteins, representing about 0.7 g/100 g d.w. (Table 1). Seeds are the vegetal part of cardoon with the highest content of proteins, ranging from 18.1–30.4% d.w. [7,35,46,47].

The most-reported proteins found in cardoon are aspartic proteases, which can be found in flowers from cardoon; they have gained special attention in dairy technology, used for milk coagulation and cheesemaking [70]. The industrial application of proteases in cheese manufacturing promotes acceptability by the vegetarian community and may improve nutritional intake [22]. The milk-clotting enzymes are known as cardosins and are isolated from cardoon flowers; nine enzymes (A–H) have been studied and characterized so far [71]. The crude extract of *C. cardunculus* flowers contains approximately 75% cardosin A and 25% cardosin B; however, cardosin A enzyme revealed a higher proteolytic activity [72]. Cardosin A is described as similar to chymosin, cleaving the same peptide bond of casein kappa [73,74]. On the other hand, cardosin B is similar to pepsin [73]. The ratio of cardosin A/cardosin B varies between thistles; for example, *Cynara humilis* only contains cardosin A [73].

#### 3.1.3. Fatty Acids and Sterols

Fatty acids are essential for human wellbeing; however, they cannot be synthesized by humans, given the lack of desaturase enzymes. For this reason, they have to be provided by the diet [75].

Table 4 presents the lipid content of vegetal parts of cardoon, as well as the total saturated fatty acids (SFAs), monounsaturated fatty acids (MUFAs), and polyunsaturated fatty acids (PUFAs), with a total of 30 fatty acids identified. In general, palmitic (C16:0; 0.03–55.9%), oleic (C18:1*n*9c; 0.58–46.6%), linoleic (C18:2*n*6c; 0.3–30.6%), and gamma-linoleic (C18:3*n*6; 0.0–70.41%) acids are the main compounds (Table 4). Seeds present the highest amounts of lipids, being also rich in MUFAs, which are useful to produce important intermediates such as azelaic and pelargonic acids. MUFAs are highly sought after by the synthetic fertilizer industry, as well as the cosmetic industry, being particularly attractive due to their low price [76,77]. 

In addition to fatty acids, vegetable oils are good sources of tocopherols and sterols. 

Phytosterols are cholesterol-like molecules found in most plant foods, and they are found in higher concentrations in vegetable oils. They can be found in free and esterified forms and can be acylated. *β*-Sitosterol, campesterol, and stigmasterol are reported as the more abundant ones in nature. Phytosterols are well recognized due to their capacity to lower blood cholesterol levels, provide protection against certain types of cancer, and possess immune-modulating activity [78]. Several sterols have been reported in several vegetal parts of cardoon but in low quantities (Table 5).

**Table 4 foods-11-00336-t004:** Fatty-acid composition of cardoon vegetal parts.

Fatty Acid (Relative Percentage, %)	Raw ^A^	Flowers ^B^	Seeds ^C^	Bracts ^D^	Heads ^E^
C6:0		n.d.–0.40	0.019–16.3	0.0094–0.05	0.082–3.710
C8:0		-	0.014–3.0	0.0086–0.0429	0.057–1.314
C10:0		-	0.015–0.151	0.00122–0.044	0.186–0.473
C11:0		0.85–1.98	0.027–0.67	0.025–0.092	0.16–0.579
C12:0		-	0.015–0.18	0.0073–0.2502	0.326–2.57
C13:0		1.05–1.17	-	n.d.–0.0105	0.0–0.084
C14:0	0.11	1.08–1.15	0.103–0.59	0.037–0.335	0.58–2.69
C14:1		-	-	n.d.–0.0083	0.0–0.54
C15:0		-	0.032–0.125	0.016–0.092	0.0–0.48
C15:1		-	-	0.0026–1.5197	0.0–1.36
C16:0	0.009–10.8	40.87–45.45	11–55.948	0.0367–6.183	14.62–43.8
C16:1	n.d.–0.02	n.d.–1.46	0.02–0.78	0.0076–0.039	0.317–12.76
C17:0	0.06	-	0.062–0.33	0.0184–0.0915	0.313–0.779
C18:0	0.002–3.6	4.83–6.28	3.289–16.39	0.209–1.775	2.687–6.0
C18:1*n*9c	0.018–27.3	2.09–3.33	0.58–3.77	0.195–0.903	4.48–46.6
C18:2*n*6c	0.041–56.8	25.10–29.91	1.7–83.3	0.054–2.208	0.748–30.6
C18:3*n*6	0.17	3.67–7.12	60.15–70.41	-	0.0–0.176
C18:3*n*3		-	0.037–4.060	0.0076–0.543	0.3675–7.5
C20:0	0.37	3.10–3.76	0.112–1.8	0.0049–0.3693	0.377–3.225
C20:1	0	-	0.08–0.49	0.0042–0.032	0.0–4.52
C20:2		-	0.18–0.24	n.d.–0.022	0.0–0.31
C21:0		-	0.064–0.079	n.d.–0.07	0.070–0.324
C20:3*n*6		-	0.012–0.015	n.d.–0.1018	0.0–8.6
C20:3*n*3		-	-	n.d.–0.156	0.0–1.38
C22:0	0.12	-	-	0.063–0.447	0.0–2.6365
C22:1	0.15	-	0.103–0.19	0.0036–0.032	0.12–4.9505
C20:5*n*3		-	-	0.0039–0.06	0.0–0.6285
C22:2		1.58–2.88	-	n.d.–0.669	0.0–0.30
C23:0		1.32–1.77	0.17–1.51	n.d.–0.06	0.26–1.61
C24:0	0.19	-	0.199–0.66	n.d.–0.128	0.0–7.411
Total variation					
SFA	0.861–15.27	57.07–64.99	15.5–95.5	1.074–9.167	22.9–61.9
MUFA	0.168–27.47	3.33–3.55	0.84–19.39	0.242–2.034	5.61–52.1
PUFA	0.211–56.8	31.66–38.61	1.9–83.7	0.076–2.955	1.895–47.7

Fatty acids are expressed as a relative percentage of each fatty acid. n.d.—not detected; C6:0—caproic acid; C8:0—caprylic acid; C10:0—capric acid; C11:0—undecanoic acid; C12:0—lauric acid; C13:0—tridecanoic acid; C14:0—myristic acid; C14:1—tetradecanoic acid; C15:0—pentadecanoic acid; C15:1—pentadecanoic acid; C16:0—palmitic acid; C16:1—palmitoleic acid; C17:0—heptadecanoic acid; C18:0—stearic acid; C18:1*n*9—oleic acid; C18:2*n*6c—linoleic acid; C18:3*n*6-gamma-linolenic acid; C18:3*n*3—alpha-linolenic acid; C20:0—arachidic acid; C20:1—gondoic acid; C20:2—eicosadienoic acid; C21:0—heneicosanoic acid; C20:3*n*6—eicosatrienoic acid; C20:3*n*3—11,14,17-eicosatrienoic acid; C22:0—behenic acid; C22:1—erucic acid; C20:5*n*3—eicosapentaenoic acid; C22:2—docosadienoic acid; C23:0—tricosanoic acid; C24:0—lignoceric acid; SFA—saturated fatty acid; MUFA—monounsaturated fatty acid; PUFA—polyunsaturated fatty acid; n-6/n-3: ratio of omega 6/omega 3 fatty acids. -, There is no data. ^A^ [48,61]; ^B^ [45]; ^C^ [7,46,47]; ^D^ [43]; ^E^ [7].

**Table 5 foods-11-00336-t005:** Phytosterol and tocopherol composition of cardoon vegetal parts.

Tocopherols (mg/100 g d.w.)	Stalks ^A^	Leaves ^B^	Flowers ^C^	Seeds ^D^	Bracts and Receptacle ^E^	Heads ^F^
*α*-Tocopherol	n.d.	39.9–100.7	n.d.	1.210–29.620	n.d.–0.062	0.25–0.619
*γ*-Tocopherol	-	-	-	n.d.	n.d.–0.120	n.d.
Cholesterol	1.0–1.3	27.6	-	-	n.d.	-
24-Methylenecholesterol	0.7–1.7	n.d.-19.3	-	5.4–6.5	n.d.	-
Campesterol	2.6–5.6	15.1–24.8	-	15.0–17.0	8.1–11.7	-
Stigmasterol	12.9–32.4	33.8–58.8	45.9–46.1	-	52.3–54.2	-
*β*-Sitosterol	13.1–25.7	63.9–171.6	49.8–70.8	-	39.2–63.7	-
*β*-Sitostanol	2.5–4.0	n.d.	n.d.	-	nd-10.6	-
Δ5-Avenasterol	n.d.	20.0–32.5	23.9–28.0	-	n.d.	-

n.d., not detected; -, there are no data. ^A^ [44];^B^ [44]; ^C^ [44]; ^D^ [46,47];^E^ [43,44]; ^F^ [7].

Tocopherols are a group of fat-soluble antioxidants and can be divided into *α*-, *β*-, *γ*-, and *δ*- forms; on the other hand, sterols constitute a major portion of the unsaponifiable fraction of oils [79,80]. The crucial effects of tocopherols and sterols toward oxidative reactions are very well documented [80].

Tocopherols are a vital fat-soluble antioxidant due to their nutritional importance, found in the leaves, seeds, bracts, and heads of cardoon. Two tocopherols (*α*-tocopherol, *γ*-tocopherol) and seven sterols are described in cardoon stalks, leaves flowers, seeds, bracts, receptacles, and heads, namely, cholesterol, 24-methylenecholesterol, campesterol, stigmasterol, *β*-sitosterol, *β*-sitostanol, and Δ^5^-avenasterol [7,43,44,46,47]. Stigmasterol and *β*-sitosterol are the majority compounds with contents ranging from 12.9 to 58.8 mg/100 g d.w. and 13.1 to 70.8 mg/100 g d.w., respectively. Leaves are the richest vegetal part of cardoon in terms of sterols. Leaves are also the vegetal part of cardoon richest in *α*-tocopherol (39–9 to 100.7 mg/100 g d.w.). In seeds, *α*-tocopherol varies between 1.210 and 29.620 mg/100 g d.w. [46,47]. In bracts and receptacles, *α*-tocopherol (n.d.–0.062 mg/100 g d.w.) and *γ*-tocopherol (n.d.–0.120 mg/100 g d.w.) can also be found [43]. In heads, amounts of *α*-tocopherol were reported to range from 0.25–0.619 mg/100 g d.w. [7].

#### 3.1.4. Organic Acids

Organic acids are primary metabolites present in high amounts in all plants, especially in fruits and vegetables. These compounds have a strong influence on the organoleptic characteristics of fruits and vegetables, especially with regard to flavor, color, and aroma [81].

Five organic acids were reported in the seeds, bracts, and heads of cardoon [7,43,46,47], namely, oxalic, quinic, and malic acids, followed by citric and fumaric acids which were detected in lower amounts. Bracts and heads were the plant parts with the highest acid contents ranging from 1.96 to 15.6 and 0.89 to 15.7 g/100 g d.w., respectively (Table 6).

**Table 6 foods-11-00336-t006:** Organic acids composition of cardoon vegetal parts.

Organic Acid (g/100 g d.w.)	Seeds ^A^	Bracts ^B^	Heads ^C^
Oxalic	0.079–0.304	0.093–9.5	0.324–12.1
Quinic	tr–0.07	tr–4.82	0.017–3.3
Malic	tr–0.086	tr–1.87	n.d.–2.31
Citric	n.d.–0.33	n.d.–1.9	n.d.–0.86
Fumaric	tr	n.d.–0.0076	n.d.–0.0542
Total variation	0.03–6.54	1.96–15.6	0.89–15.7

d.w., dry weight; tr, traces; n.d., not detected. ^A^ [46,47]; ^B^ [43]; ^C^ [7].

### 3.2. Micronutrients

Micronutrients are essential elements required by the organism to maintain a healthy status. The requirement for micronutrients differs depending on the individual need, which varies due to the different metabolic conditions within the life cycle (age, lifestyle, hormonal activity, exercise, etc.) [82]. They cannot be synthesized within the body, being all the essential micronutrients supplied via our diet [83].

#### 3.2.1. Minerals

Mineral micronutrients are involved in several biochemical processes, such as controlling blood pressure and decreasing the risk of cardiovascular problems; an adequate intake of these chemical compounds is essential to the prevention of deficiency-related diseases [83]. For example, calcium is crucial for the human body, particularly for bone mass; on the other hand, an inadequate intake of potassium and/or sodium is related to the development of hypertension [84].

A total of seven minerals have been reported in raw cardoon, as well as the flowers and seeds of cardoon, namely, K, Na, Ca, Mg, Mn, Fe, and Zn [45,47]. Considerable amounts of K, Ca, Mg, and Fe have been reported in seeds, while the Na level is considerably low (12 to 24 mg/100 g of d.w.) [47], being very important from a nutritional point of view. On the other hand, raw cardoon and flowers present higher amounts of Na (170 and 200–300 mg/100 g of d.w., respectively) [45,48]. Flowers also present high amounts of K (1600–1710 mg/100 g of d.w.), Ca (580–1010 mg/100 g of d.w.), and Mg (150–170 mg/100 g of d.w.) (Table 7) [45].

**Table 7 foods-11-00336-t007:** Mineral composition of cardoon vegetal parts.

Minerals (mg/100 g d.w.)	Raw ^A^	Flowers ^B^	Seeds ^C^
K	400	1600–1710	493–880
Na	170	200–300	12–24
Ca	70	580–1010	734–1583
Mg	42	150–170	241–809
Mn	0.256	-	4.21–6.2
Fe	0.7	-	9.2–16
Zn	0.17	-	1.4–5.2

-, there are no data. ^A^ [48]; ^B^ [45]; ^C^ [47].

#### 3.2.2. Vitamins

Plants are a major source of vitamins, which maintain health in humans and animals, as well as play important roles in plant growth and development. Vitamins include two major groups: fat-soluble (A, D, E, and K) and water-soluble vitamins (thiamine (B1), riboflavin (B2), niacin (B3), pantothenic acid (B5), pyridoxine (B6), biotin (B7), folate (B9), cobalamin (B12), and C families) [85]. Despite being minority compounds found in fruits, vegetables, and other foods, vitamins are essential for the maintenance and function of the organism and normal growth [86]. Cardoon has several vitamins reported in its raw material, contributing to its bioactive effects, e.g., antioxidant activity, decreasing the loss of vision and blurry vision, and supporting normal growth [85,86]. Vitamin C is the most abundant vitamin in cardoon and artichoke, with contents of 2.0 and 11.7 mg/100 g raw edible portion, respectively, followed by pantothenic acid (0.338 and 0.34 mg/100 g raw edible portion, respectively) and niacin (0.3 and 0.9 mg/100 g raw edible portion, respectively). Other vitamins reported in cardoon and artichoke are thiamine, riboflavin, pyridoxine, folate, vitamin A, and vitamin E (Table 8).

Vitamin C is an essential nutrient which cannot be synthesized by humans due to the lack of a key enzyme in the biosynthetic pathway. Vitamin C is crucial in the metabolism of proteins and the biosynthesis of some neurotransmitters [87,88]. Ongoing research is studying whether vitamin C, due to its antioxidant activity, can help prevent or delay the development of diseases in which oxidative stress plays an important role. 

**Table 8 foods-11-00336-t008:** Vitamin content of raw cardoon and raw artichoke.

Vitamins (per 100 g Edible Portion)	Cardoon Raw	Artichoke Raw
Vitamin C (mg)	2.0	11.7
Thiamin (mg)	0.02	0.07
Riboflavin (mg)	0.03	0.07
Niacin (mg)	0.3	0.9
Pantothenic acid (mg)	0.338	0.34
Vitamin B6 (mg)	0.116	0.12
Total folate (µg)	68.0	68.0
Vitamin A µg)	0	8.0
Vitamin E (mg)	-	0.19

Data adapted from [48]. -, There is no data.

### 3.3. Phytochemicals

Phytochemicals constitute phytonutrients, which are natural bioactive components rich in foods such as fruits, vegetables, nuts, seeds, tea, and dark chocolate. The most common phytochemicals reported in foods are phenolic compounds (non-flavonoids and flavonoids), carotenoids, coumarins, indoles, lignans, organosulfurs, stilbenoids, isothiocyanates, saponins, phenylpropanoids, anthraquinones, ginsenosides, phytosterols, fibers, etc. [89].

#### 3.3.1. Volatile Compounds

The plant release of volatile compounds is involved in a broad class of ecological functions since volatiles play an important role in plant interactions with biotic and abiotic factors. Volatile compounds in plant materials can be derived from amino acids, fatty acids, and carbohydrates [90]. They vary widely across species and maintain differences in ecological strategy. They are released by almost any kind of tissue and type of vegetation (trees, shrubs, grass, etc.) as green leaf volatile compounds, nitrogen-containing compounds, and aromatic compounds. Plant volatiles can be emitted constitutively or in response to a variety of stimuli [91]. 

Several families of volatile organic compounds are reported in cardoon vegetal parts, namely, aromatic compounds, fatty acids, long-chain aliphatic alcohols, sesquiterpene lactones, pentacyclic triterpenes, and others (Table 9).

Sesquiterpene lactones represent one of the most prevalent and biologically relevant groups of secondary metabolites reported in plants. They are responsible for the phytotoxic, cytotoxic, fungicidal, antiviral, and antimicrobial activity of cardoon [92]. Cynaropicrin is a guaianolide sesquiterpene lactone found in the stalks, receptacle, bracts, and leaves of cardoon [44], as well as in globe artichoke [93], being an interesting compound due to its availability and great biological potential, such as antitumoral, antifeedant, and anti-inflammatory activity [44,94,95]. Grosheimin was found in higher amounts in leaves, with minor amounts in the stalks. Furthermore, deacylcynaropicrin was only reported in the leaves of cardoon, but in lower concentrations [44]. Both compounds are less studied than cynaropicrin.

Long-chain aliphatic alcohols were described as being in low concentrations, accounting for 1–5% in vegetal parts of cardoon [44]. This family of compounds has been recognized to present several biological and pharmacological effects in animals and humans. For example, octocosan-1-ol was reported to have several properties such as antioxidant activity, a cholesterol-lowering effect, lipid peroxidation, neurological functions, antiangiogenic and antitumor activity, prevention of inflammation and pain, antifatigue function, ergogenic properties, and improving athletic performance in exercise-trained rats, being used in food, medicine, and cosmetics, as reviewed by Kabir [96].

Pentacyclic triterpenes are ubiquitously distributed throughout the plant kingdom in a free form as aglycones or in combined forms, and they have long been known to have several biological effects. These compounds are the major family of lipophilic compounds found in the capitula, receptacle, and bracts (Table 9), in amounts that vary from 8% to 84% in the vegetal parts of cardoon [44]. Compounds such as *α*, *β*-amyrins have been reported to possess several pharmacological effects, such as anxiolytic, anti-depressant, anticancer, anti-inflammatory, antimicrobial, and antiviral activity [86].

Aromatic compounds and others were found in lower amounts in the vegetal parts of cardoon (contents ranging from 0.3% and 4% of the total detected compounds) [44]. 3-Vanillylpropanol was the major compound found in leaves, followed by benzoic acid. Furthermore, scopoline was only reported in the capitula and florets [44]. Aromatic compounds have been described as antioxidant and antimicrobial agents [97]. These compounds have been studied due to their advantages over antibiotics as growth promoters, being reported as residue-free and generally recognized as safe (GRAS) [98].

**Table 9 foods-11-00336-t009:** Volatile organic compounds found in vegetal parts of cardoon.

Volatile Organic Compounds	Vegetal Parts	References
Aromatic compounds		
Benzoic acid	Stalks, receptacles and bracts, florets, leaves	[44]
Vanillin	Stalks, receptacles and bracts, leaves	[44]
Syringaldehyde	Stalks, receptacles and bracts, florets, leaves	[44]
2,6-Dimethoxyhydroquinone	Stalks, receptacle and bracts, florets	[44]
3-Vanillylpropanol	Stalks, florets, leaves	[44]
Vanillylpropanoic acid	Florets	[44]
Scopolin	Florets	[44]
Benzaldehyde	Stalks, leaves	[33,99]
Furfural	Leaves	[33,99]
(*E*)-2-Hexanal	Leaves	[99]
1-Octen-3-one	Leaves	[99]
6-Methyl-5-hepten-2-one	Leaves	[99]
Octanal	Leaves	[99]
Benzene acetaldehyde	Leaves	[99]
(*E*)-2-Octenal	Leaves	[99]
Acethophenone	Leaves	[99]
(*E*,*E*)-3,5-Octadien-2-one Nonanal	Leaves	[99]
	Leaves	[99]
(*E*)-6-Methyl-3,5-heptadien-2-one	Leaves	[99]
Phenetyl alcohol	Leaves	[99]
Isophorone	Leaves	[99]
3-Nonen-2-one	Leaves	[99]
(*E*,*Z*)-2,6-Nonadienal 4-Methyl-Acephenone	Leaves	[99]
	Leaves	[99]
Safranal	Leaves	[99]
Decanal	Leaves	[99]
* β*-Ciclocitral	Leaves	[99]
Neral	Leaves	[99]
*β*-Homocyclocitral	Leaves	[99]
Geranial	Leaves	[99]
*p*-Vinylguaiacol	Leaves	[33,99]
Eugenol	Leaves	[33,99]
*γ*-Nonalactone	Leaves	[99]
(*E*)-*β*-Damascenone	Leaves	[99]
Geranyl acetone	Leaves	[99]
*β*-Ionone	Leaves	[99]
Dicyclohexyl-methanone	Leaves	[99]
Dihydroactinidiolide	Leaves	[99]
Phytone	Leaves	[99]
Fatty acids		
Saturated		[44]
Tetradecanoic acid	Stalks, receptacle and bracts, florets, leaves	[44]
Pentadecanoic acid	Stalks, receptacle and bracts, florets, leaves	[44]
Hexadecanoic acid	Stalks, receptacle and bracts, florets, leaves	[44]
Heptadecanoic acid	Stalks, receptacle and bracts, florets, leaves	[44]
Octadecanoic acid	Stalks, receptacle and bracts, florets, leaves	[44]
Nonadecanoic acid	Stalks, receptacle and bracts, florets, leaves	[44]
Eicosanoic acid	Stalks, receptacle and bracts, florets, leaves	[44]
Heneicosanoic acid	Stalks, receptacle and bracts, florets, leaves	[44]
Docosanoic acid	Stalks, receptacle and bracts, florets, leaves	[44]
Tricosanoic acid	Stalks, receptacle and bracts, florets, leaves	[44]
Tetracosanoic acid	Stalks, receptacle and bracts, florets, leaves	[44]
Pentacosanoic acid	Stalks, receptacle and bracts, florets	[44]
Hexacosanoic acid	Stalks, florets, leaves	[44]
Octacosanoic acid	Stalks, florets, leaves	[44]
Unsaturated		
*cis-*9-Hexadecenoic acid	Stalks, receptacle and bracts, florets, leaves	[44]
*trans*-9-Hexadecenoic acid	Stalks, receptacle and bracts, florets, leaves	[44]
9,12-Octadecadienoic acid	Stalks, receptacle and bracts, florets, leaves	[44]
9,12,15-Octadecatrienoic acid	Stalks, receptacle and bracts, florets, leaves	[44]
*cis*-9-Octadecenoic acid	Stalks, receptacle and bracts, florets	[44]
*trans*-9-Octadecenoic acid	Stalks, receptacle and bracts, florets	[44]
Hydroxy fatty acids	Stalks, florets	[44]
2-Hydroxyheptanoic acid	Stalks	[44]
2-Hydroxyundecanoic acid	Florets	[44]
Long-chain aliphatic alcohols		
Hexadecanol-1-ol	Stalks, receptacle and bracts, florets, leaves	[44]
*cis*-9-Octadecen-1-ol	Stalks, receptacle and bracts, florets	[44]
Octadecan-1-ol	Stalks, receptacle and bracts, florets, leaves	[44]
Eicosan-1-ol	Stalks, receptacle and bracts, florets	[44]
Docosan-1-ol	Stalks, receptacle and bracts, florets, leaves	[44]
Tetracosan-1-ol	Stalks, receptacle and bracts, florets, leaves	[44]
Hexacosan-1-ol	Stalks, receptacle and bracts, florets, leaves	[44]
Octocosan-1-ol	Stalks, florets, leaves	[44]
Sesquiterpene lactones		
Grosheimin	Stalks, leaves	[44]
Deacylcynaropicrin	Stalks, receptacle and bracts, leaves	[44]
Cynaropicrin	Stalks, receptacle and bracts, leaves	[44]
Pentacyclic triterpenes		
*β*-Amyrin	Stalks, receptacle and bracts, florets, leaves	[44]
*α*-Amyrin	Stalks, receptacle and bracts, florets, leaves	[44]
*β*-Amyrin-acetate	Stalks, receptacle and bracts, florets, leaves	[44]
*α*-Amyrin acetate	Stalks, receptacle and bracts, florets, leaves	[44]
Lupenyl acetate	Stalks, receptacle and bracts, florets, leaves	[44]
*Ψ* -Taraxasterol	Stalks, receptacle and bracts, florets, leaves	[44]
Taraxasterol	Stalks, receptacle and bracts, florets, leaves	[44]
*Ψ* -Taraxasterol acetate	Stalks, receptacle and bracts, florets, leaves	[44]
Taraxasteryl acetate	Stalks, receptacle and bracts, florets, leaves	[44]
Others		
Inositol	Stalks, receptacle and bracts, florets, leaves	[44]
2,3-Dihydroxypropyl hexadecanoate	Stalks, receptacle and bracts, florets, leaves	[44]
*trans*-Squalene	Leaves	[44]

#### 3.3.2. Phenolic Compounds (Flavonoids and Non-Flavonoids)

The consumption of medicinal plants, fruits, and vegetables is associated with a wide array of benefits in human health, linked to the consumption of non-nutrient bioactive compounds, such as phenolic compounds, for which various biological activities have been described, such as antioxidant, anti-inflammatory, regulator of glucose levels, antimutagenic, anticancer, and neuroprotective potential [5,30,100].

Several phenolic compounds have already been reported in plants (>10,000 phenolics) [101]. They are divided according to their chemical structure into non-flavonoids (phenolic acids, stilbenes, and lignans) and flavonoids (flavonols, flavan-3-ols, flavones, flavanones, and anthocyanidins) [86].

The composition of phenolic compounds in cardoon, wild cardoon, and globe artichoke is associated with several factors, such as different cultivars, genotypes, growing conditions, edaphoclimatic conditions, plant parts, plant age, harvest time factors, and handling and storage factors [7,30,31,43,46,102,103].

Cardoon presents mono- and di-isomers of caffeoylquinic acid, such as 1,5-dicaffeoylquinic acid (cynarine) and flavonoid *O*-glycosides (luteolin and apigenin derivatives) as the main compounds responsible for its biological effects [13,102,104]. Cynarine has been greatly investigated due to its high capacity to inhibit cholesterol biosynthesis and LDL oxidation, consequently reducing the risk of cardiovascular diseases [37]. Additionally, the anti-obesity effect of chlorogenic acid was reported in mice [105]. Furthermore, several reports also showed further biological potential of artichoke and cardoon extracts, rich in phenolic compounds, such as antimicrobial [106,107,108,109], anti-inflammatory [110], and antitumor properties [111,112].

In general, the compounds mostly detected in cardoon leaves and heads are chlorogenic acid and 1,5-*O*-dicaffeoylquinic acid [7,31,102,113,114]. Immature heads revealed a higher phenolic content compared with mature ones [7,31], proving that the maturity stage possesses a significant influence on the phenolic profile [31,115]. Apigenin derivatives were reported in the highest concentrations in both wild and cultivated cardoon immature inflorescences [116]. Furthermore, high contents of hydroxycinnamic acids, mainly dicaffeoylquinic acids, in receptacle and bracts, and apigenin and luteolin derivatives, in capitulum florets and leaves, respectively, were reported in different plant tissues of cardoon [114] (Table 10).

Several studies reported the phenolic composition of artichoke, with caffeic acid derivatives identified as the major class of phenolics. 5-*O*-Caffeoylquinic acid is the most abundant, followed by 1,5-*O*-dicaffeoylquinic acid, and 3,4-*O*-dicaffeoylquinic acid, while it is also rich in flavonoids, with luteolin-7-O-glucoside being the major flavonoid reported [9,17]. These compounds were also found in higher amounts in raw artichoke and in their vegetal parts, as seen in cardoon vegetal parts [7,9,17,30,31,43,46].

## 4. *Cynara cardunculus* and Its Effect on Metabolic Disorders

Metabolic disorders are a common problem in industrialized countries, associated with an unhealthy lifestyle, such as a sedentary routine, high levels of stress, and unhealthy eating habits [119]. These disorders are related to several factors, such as obesity, diabetes, hypertension, and dyslipidemia [120].

According to the WHO, more than 100 million Europeans rely on traditional and complementary medicine [121]. Plants have been widely described as having activity against several metabolic disorders. Their activity against these disorders is linked to their composition in several bioactive compounds, which, via various mechanisms of action, are related to the regulation of oxidative processes, improving the antioxidant activity, decreasing the production of proinflammatory cytokines, and interacting in intracellular signaling pathways [122,123].

Thus, *C. cardunculus* L. is one of the plants that has been described in the literature with therapeutic purpose [124]. Its extracts have been used in traditional medicine, due to their hepatoprotective (Table 11), lipid-lowering (Table 12), and antidiabetic (Table 13) activities. Furthermore, *C. cardunculus* L. extracts possess antimicrobial [125,126,127], antioxidant [125,127,128], and anti-inflammatory [127,129] properties, as extensively described in the literature. The possible mechanism supporting the biological activity of *C. cardunculus* might be due to its content of vitamins, flavonoids, and polyphenols, since they are potent free-radical scavengers that prevent oxidative stress, which is intrinsically connected to the hepatoprotective, hypolipidemic, and antidiabetic activity [130,131].

### 4.1. Hepatoprotective Activity

The liver plays a relevant role in a variety of regulatory and metabolic activities; some of its functions influence the metabolism of carbohydrates, lipids, and proteins, while this organ is also crucial in endocrine activity and the elimination of toxins from the organism [132]. However, liver functions can be easily compromised, since hepatic damage can be triggered by prescription drugs, over-the-counter medications, dietary supplements, environmental factors, and alcohol [133].

Researchers have been exploring the hepatoprotective effect of *C. Cardunculus* taxa. Several studies demonstrated through in vitro assays, *in vivo* assays, and clinical trials that *C. cardunculus* L. (wild cardoon, domesticated cardoon, and globe artichoke) possesses hepatoprotective activity, as summarized in Table 11.

The in vitro hepatoprotective activity of *C. scolymus* on a human hepatocellular carcinoma cell line (HepG2 cells) was demonstrated by Youssef and colleagues [134], with the synergic activity resulting from the combination of three extracts (*C. cardunculus* L. var. *scolymus* L. Fiori, *Ficus carica* L. (fig), and *Morus nigra* L. (blackberry)) which were able to normalize the enzyme levels in the cells after the induction of hepatoxicity using carbon tetrachloride (CCl4), which caused an increase in aspartate aminotransferase (AST) and alanine aminotransferase (ALT) levels and a reduction in glutathione (GSH) and superoxide dismutase (SOD). 

Furthermore, El Morsy and collaborators [135] studied the potential protective effect of artichoke leaf extract (ALE) against hepatotoxicity triggered by paracetamol overdose in rats. The authors observed that rats pretreated with ALE had lower values of serum ALT and AST, which were previously raised by paracetamol. Furthermore, hepatotoxicity due to paracetamol also causes oxidative damage. For this reason, the ALE effect on enzymatic antioxidants was also studied, and the authors found that ALE increased hepatic GSH, which reversed oxidative stress parameters, DNA damage, and necrosis. Moreover, it is important to emphasize that ALE had a higher impact on the restoration of GSH, nitric oxide, and antioxidant enzyme levels compared to *N*-acetylcysteine, a reference drug for the treatment of paracetamol overdose [135]. In a similar study, Elsayed Elgarawany and colleagues [136] investigated the effect of artichoke leaves, silymarin, and both compounds in acetaminophen-induced liver damage in mice. The treatment with artichoke, silymarin, or their combination resulted in a significant decrease in liver weight and enzymes. However, the reduced expression of proliferative cell nuclear antigen happened in the group using ALE and silymarin, which was attributed to the capacity of silymarin to increase the activity of glutathione reductase [136].

On the other hand, Tang and colleagues evaluated the effect of ALE on the alcoholic liver disease animal model; results showed that the levels of hepatic enzymes were reduced, and there was a suppression of the Toll-like receptor (TLR) 4 and nuclear factor-kappa B (NF-*κ*B) pathway, which ultimately mitigated the oxidative stress and the inflammatory pathway [137].

Additionally, the hepatoprotective effect of bergamont polyphenolic fraction (*Citrus bergamia*) and *C. cardunculus* var. *scolymus* L. Fiori extract was also studied in a clinical trial conducted by Musolino and collaborators. After 16 weeks of supplementation, both extracts alone (300 mg/daily) and in combination (150 mg/kg of each extract) demonstrated the potential to decrease serum ALT and AST; nevertheless, the combination of both extracts exhibited better results [138].

Some mechanisms of action of artichoke extract have been described as being related to the ability to influence the lipid profile, leading to a hepatoprotective effect. For this reason, one of the mechanisms of action underlying the hepatoprotective activity might be related to the suppression of the TLR 4 and NF-*κ*B pathway, which can ultimately hinder the release of proinflammatory cytokines and subsequently prevent the damage of the cell membrane, thus avoiding the leakage of AST and ALT into the bloodstream [139]. The bioactive compounds of *Cynara* that might be responsible for this hepatoprotective activity are caffeic acid and its derivates (chlorogenic acid and cynarine), as well as luteolin [139,140,141], mainly attributed to the antioxidant properties of the above-stated components [141].

**Table 11 foods-11-00336-t011:** Overview of the hepatoprotective activity of *Cynara Cardunculus*.

Plant Variety	Intervention	Study Type	Model	Results	Ref.
var. *scolymus* L. Fiori	Aqueous infusion of flower heads	*in vitro*	CCl_4_-induced damage in HepG2 cells	The aqueous infusion of separate flower heads of *C. scolymus*, fruit of *Ficus carica* L., and fruit of *Morus nigra* L. presented reduced levels of liver enzymes and a higher level of antioxidant enzymes. The mixture of the three plants decreased AST and ALT levels and increased GSH and SOD when compared to CCl_4_-treated cells.	[134]
	Ethanol extract from leaves	*in vivo*	Wistar Rats with high-fat diet (HFD)-induced obesity	The administration of artichoke extract caused a decrease in pancreatic lipase compared to the HFD group and reduced organ weight. In addition, the serum lipid profile and the values of the hepatic enzymes were restored.	[142]
	Hydroethanolic extract from leaves	*in vivo*	Wistar Rats with phenylhydrazine-induced hemolytic anemia	The groups treated with *Cynara scolymus* extracts exhibited a decrease in the serum liver enzymes levels and, consequently, an improvement of the liver tissue damage.	[143]
	Ethanolic extracts from artichoke (plant part)	*in vivo*	ICR mice with acute alcohol-induced liver injury	The pretreatment with 1.6 g/kg BW of artichoke had a preventive effect due to its ability to reduce the lipidic profile and MDA while increasing SOD and GSH, as well as the inhibition of the inflammatory pathway by suppressing the expression levels of TLR4 and NF-*κ*B.	[137]
	Methanol extract from leaves	*in vivo*	Mice infected with *Schistosoma mansoni*	Upon treatment with ALE either alone or in conjugation with praziquantel, there was an improvement of the liver enzymes. In addition, ALE-treated groups exhibited a reduction in the granuloma size due to the increase in hepatic stellate cell recruitment, ultimately improving liver fibrosis.	[144]
	Aqueous leaf extract	*in vivo*	High-fat and high-cholesterol diet-induced steatohepatitis and liver damage in mice	After ALE treatment, there was a reduction of the hepatic triglyceride level and inflammation, as well as a suppression of the liver damage induced by high-fat and high-cholesterol diet in mice.	[145]
var. *scolymus* L. Fiori	Ethanol extract from receptacle, stem, inner bract, outer bract, and leaves	*in vivo*	Sprague–Dawley rats with paracetamol-induced hepatotoxicity and nephrotoxicity	There were no significant changes in ALT and AST levels after the treatment with the different parts from artichoke; however, the histopathological data showed that the receptacle and stem extracts of *Cynara scolymus* significantly improved the pathological changes induced by paracetamol in both organs.	[146]
var*. scolymus* L. Fiori	Aqueous extract from heads and leaves	*in vivo*	Sprague–Dawley rats with diethylnitrosamine (DEN)-induced hepatocellular carcinoma	The treatment with fish oil 10% or 1 g of artichoke leaves led to better improvement of DEN-induced changes in the biochemical parameters, such as antioxidant enzymes, angiogenic growth factors, liver function enzymes, and other substances produced by the liver.	[147]
	Ethanol extract from leaves	*in vivo*	Albino mice of C57BL/6 with acetaminophen-induced hepatotoxicity	After the treatment with ALE, silymarin, or their conjugation, there was an improvement of the serum liver enzymes, a decrease in MDA levels, an increase in glutathione reductase, and a decrease in PCNA expression. Thus, both plant extracts had hepatoprotective activity against acetaminophen.	[136]
	Hydroethanolic extract from leaves	*in vivo*	Wistar rats with acutediazinon-induced liver injury	The treatment with ALE caused a reduction in serum ALP, AST, ALT, MDA, TNF-*α*, and protein carbonyl and enhanced liver histopathological changes and hepatic CAT and SOD activities.	[130]
	Aqueous extract from leaves	*in vivo*	Sprague–Dawley rats with paracetamol-induced hepatotoxicity	Pretreatment with ALE (1.5 g/kg) reduced ALT and AST levels. In addition, the hepatic lipid peroxidation and NO levels also exhibited a reduction, whereas SOD and antioxidant enzymes activity increased. Furthermore, the pretreated group presented a reduction in DNA damage.	[135]
	Ethanol extract from leaves	*in vivo*	Wistar Rats with cadmium (Cd) toxicity-induced oxidative organ damage	The ALE-treated rats exhibited a significant reduction in the oxidative stress in the Cd-exposed rats. Furthermore, ALE alone increased the GSH and CAT activity levels in rat liver and reduced liver lesions.	[148]
var. *scolymus* L. Fiori	Ethanol extract from leaves	*in vivo*	Sprague–Dawley rats with carbon tetrachloride-induced oxidative stress and hepatic injury	After treating rats with ALE, the levels of AST and ALT were decreased by 40% and 52%, respectively, in the curative group compared to the CCl_4_ group, normalized to the antioxidant system, due to the decrease in SOD and MDA levels.	[149]
	Luteolin-enriched artichoke leaves	*in vivo*	C57BL/6N mice with HFD-induced obesity	Luteolin-enriched artichoke leaves and luteolin treatments significantly decreased the hepatic PAP enzyme activity and increased the activity of hepatic CPT.	[150]
	Hydroethanolic extract of artichoke	*in vivo*	Wistar rats with lead acetate-induced toxicity	Upon the treatment with the ALE, there was a decrease in the lipid profile levels and lead serum levels.	[151]
var. *altilis* DC.	Aqueous extract from leaves	*in vivo*	Rats with 2,4,6-trinitrobenzenesulfonic acid (TNBS)-induced colitis	After treating the mice with the extract, there was no significant variation in the ALT values comparing with the TNBS-treated mice.	[152]
	Butanolic extract from aerial parts	*in vivo*	Male albino rats with paracetamol induced liver injury	After 10 days, rats pretreated with the extract (300 mg/kg) prevented the increase in AST, ALT, and ALP caused by paracetamol.	[153]
var*. scolymus* L. Fiori	Hydroethanolic Extract from Leaves	*in vivo*	Liver damage induced by deltamethrin in weanling male rats	After 28 days of supplementation with an herbal syrup (sucrose, extract from C. *scolymus*, and chicory), there was an increase in the oxidative stress enzymes and an improvement of the serum liver enzymes.	[154]
var*. altilis* DC.	Leaves extract encapsulated in capsules	Clinical trial	Adult patients with a history of at least 12 months of T2DM and NAFLD	After 16 weeks, the patients supplemented with C. *altilis* alone exhibited a reduction in AST and ALT levels. However, the combination of *Citrus bergamia* and C. *altilis* had a more significant reduction, leading to liver protection.	[138]
var. *scolymus* L. Fiori	Cynarol^®^ (Leaves extract)	Clinical trial	Subjects with nonalcoholic fatty liver disease	After the treatment with ALE the hepatic vein flow increased, and portal vein diameter and liver size decreased when compared with placebo. Furthermore, the treatment also reduced both AST and ALT, as well as traditional liver markers and serum lipid profile.	[155]
	-	Clinical trial	Patients with nonalcoholic steatohepatitis	After the treatment with *C. scolymus* L. extract, there was a decrease in ALT and AST, while it also reduced the serum lipid profile.	[141]

-, there are no data.

### 4.2. Hypolipidemic Activity

Hyperlipidemia is characterized as an abnormally high concentration of lipids in the blood, such as triglycerides (TG), total cholesterol (TC), and low-density lipoprotein cholesterol (LDL-c) and a reduction in high-density lipoprotein cholesterol (HDL-c) values [156]. This unbalanced lipidic profile is one of the most relevant risk factors for cardiovascular diseases [157]. The most common causes of hyperlipidemia are lifestyle habits (smoking, obesity, and sedentarism), type 2 diabetes, alcohol, lipoprotein lipase mutations, hypothyroidism, and environmental and genetic factors [158]. The beneficial effects of *C. cardunculus* on the lipidic profile have been the focus of several studies, as described in Table 12. 

Ben Salem and collaborators demonstrated the effect of ALE on the lipidic profile, cardiac markers, and antioxidant levels in obese rats [159]. As a result of supplementation of rats with ALE (200 mg/kg and 400 mg/kg), there was an improvement in the lipidic profile due to the decrease in TC, TG, and LDL-c levels and increase in HDL-c levels. In addition, ALE supplementation also reduced the cardiac markers and increased the antioxidant enzyme SOD, GPx, and GSH activities [159]. The effect of artichoke extracts on liver phosphatide phosphohydrolase and lipid profile in hyperlipidemic rats was also evaluated by Heidarian and colleagues. The supplementation with a mix of 10% artichoke in rat pellets for 60 days resulted in a decrease in the lipids in the serum and the phosphatide phosphohydrolase activity, which consequently decreased the levels of TG [151]. Furthermore, Oppedisano and collaborators showed the effect of *C. cardunculus* var. *sylvestris* (Lank) Fiori leaf extract in rats fed an HFD. In the animals fed an HFD and supplemented with 10 or 20 mg/kg of *C. cardunculus*, there was a dose-dependent reduction in TC, TG, and malondialdehyde levels compared with the group that was not supplemented with *C. cardunculus* var. *sylvestris* (Lank) Fiori [160].

Furthermore, TC, LDL-c/HDL-c and TC/HDL-c ratio levels suffered a significant reduction in subjects with primary mild hypercholesterolemia and after 8 weeks of supplementation with ALE (2 daily doses of 250 mg). There was a significant increase in HDL-c, which plays an important role in the prevention of cardiovascular diseases [161]. 

Luteolin and chlorogenic acid are compounds with crucial roles in modulating the lipid profile, according to several authors [139,159]. Luteolin can reduce cholesterol synthesis owing to the inhibition of hydroxy-methyl-glutaryl-coenzyme A (HMG-CoA) reductase, due to the inhibition of liver sterol regulatory element-binding proteins (SREBPs) and acetyl-CoA C-acetyltransferase (ACAT) [139,162]. However, the modulation of the lipidic profile by luteolin is not clear. Kwon and colleagues evaluated the effect of luteolin-enriched artichoke versus artichoke leaf and concluded that artichoke leaf had a higher effect on the lipidic profile than the luteolin-enriched [150]. 

Chlorogenic acid stimulates AMP-activated protein kinase and consequently inhibits sterol regulatory element-binding protein, which results in a reduction in cholesterol synthesis. Additionally, chlorogenic acid can also increase *β*-oxidation and inhibit malonyl-CoA, due to carnitine palmitoyl transferase stimulation, which subsequently lowers triglycerides levels [139,141]. Lastly, other authors also mentioned inulin, capable of modulating the lipid profile by increasing the conversion of cholesterol in bile salts, subsequently reducing very-low-density lipoprotein and LDL-c serum levels [139,163].

**Table 12 foods-11-00336-t012:** Overview of the hypolipidemic activity of *Cynara cardunculus*.

Variety/Species	Intervention	Study Type	Model	Result	Ref.
var. *scolymus* L. Fiori	Ethanol extract from Leaves	*in vivo*	HFD-induced cellular obesity and cardiac damage in Wistar Rats	Oral administration of ALE at two doses 200 and 400 mg/kg decreased lipase pancreatic activity, improved the lipid profile, significantly decreased the cardiac markers, and improved the antioxidant activity and oxidative stress markers.	[159]
	Ethanol extract from leaves	*in vivo*	HFD-induced obesity in Wistar rats	After the treatment, there was a normalization of serum lipid profile, a decrease in urea, uric acid, and creatinine, a reduction in the formation of lipid accumulation and AOPP, and an increment in the antioxidant activity.	[164]
	Aqueous extract from Leaves	*in vivo*	Golden Syrian hamsters	After 42 days, the ALE fed hamsters exhibited a decrease in TC, non-HDL, and TG. In addition, there was an increase in the excretion of bile acids and neutral sterols.	[165]
	Aqueous extract from Leaves	*in vivo*	Wistar rats fed on high-cholesterol diet	The rats fed on high-cholesterol diet treated with ALE exhibited a decrease in the serum lipidic profile.	[166]
	Extract from Leaves	*in vivo*	C57BL/6N mice with HFD-Induced Obesity	When treated with ALE and luteolin-enriched ALE, there was a decrease in the plasma, hepatic, and fecal lipid profile, as well as a decrease in the hepatic PAP enzyme activity.	[150]
var. *sylvestris (*Lank) Fiori	Hydroalcoholic extract from leaves	*in vivo*	Insulin resistance, hyperlipidemia, and NAFLD in Sprague–Dawley rats fed a hyperlipidemic diet	After the treatment with ALE, the rats on a high-fat diet exhibited a decrease in the serum lipid profile in a dose-dependent manner.	[160]
var. *scolymus* L. Fiori	Ethanol extract from leaf	*in vivo*	DNA damage and atherosclerosis in Wistar albino rats fed an atherogenic diet	TG and HDL-c showed a significant decrease in the ALE-treated rats, when compared to the control group.	[167]
var. *scolymus* L. Fiori	Aqueous extract from leaves	*in vivo*	Hepatic and cardiac oxidative stress in Wistar rats fed on high-cholesterol diet	Hypercholesterolemic rats treated with ALE exhibited a decrease in the serum TC and TG levels; however, in the liver, there was no change. On the other hand, this treatment reduced the MDA and diene conjugate levels in the liver and heart tissues, and there was an increase in hepatic vitamin E levels and GSH-Px activities.	[168]
	Aqueous extract from leaves and aqueous extract from stems	*in vivo*	Rats fed with HFD and vitamin C supplement	Both extracts reduced the lipid profile levels, as well as GOT and GTP values.	[169]
Red yeast rice extract, policosanol, and var. *scolymus* L. Fiori	Limicol^®^ (w/200 mg of *Cynara scolymus* L. leaf extract)	*in situ* and clinical trial	Wistar rats and subjects with untreated hypercholesterolemia	After 4 weeks of supplementation, LDL-c and TC were significantly lower in the supplemented group than in the placebo.	[170]
var. *scolymus* L. Fiori	Hydroalcoholic extract from leaves	Clinical trial	Women with metabolic syndrome	After 12 weeks of treatment with ALE, the carriers of A allele of FTO-rs9939609 exhibited a significant decrease in serum TG level compared with the controls. However, there was no genotype–intervention interaction for the TCF7L2-rs7903146 polymorphism.	[171]
Red yeast rice extract, policosanol and var. *scolymus* L. Fiori	Limicol^®^ (w/200 mg of *C. scolymus* Leaf extract)	Clinical trial	Subjects with moderate hypercholesterolemia	LDL-c and TC were reduced by, respectively, 21.4% and 14.1% at week 16 in the supplemented group compared with baseline. Furthermore, triglyceride levels decreased by 12.2% after 16 weeks in the supplemented group.	[172]
			Subjects with moderate untreated hypercholesterolemia	The group treated with a supplement of red yeast rice, policosanols, and artichoke leaf extracts exhibited a significant decrease in LDL-c, TC, and apo B in healthy subjects with moderate hypercholesterolemia. On the other hand, no effect was demonstrated on other lipid concentrations, and there was no alteration in liver and renal function markers and in the muscle breakdown.	[173]
var. *scolymus* L. Fiori	Extract from leaves	Clinical trial	Patients with nonalcoholic steatohepatitis	After the treatment with ALE, the AST and ALT levels decreased, as well as the TG and TC levels.	[141]
	Extract from leaves	Clinical trial	Subjects with primary mild hypercholesterolemia	The supplemented group exhibited a significant decrease in TC, LDL-c, and LDL/HDL compared with the placebo. Furthermore, the supplementation caused an increase in HDL-c, which might have clinical interest owing to its protective role in cardiovascular disease.	[161]

### 4.3. Antidiabetic Activity

Diabetes mellitus (DM) is described as having abnormal blood glucose levels caused by a defect in insulin secretion and/or insulin action. These abnormal glucose levels are associated with lipidic profile dysregulation [174,175]. Several authors have researched the antidiabetic activity of C. *cardunculus* and its possible mechanisms of action (Table 13).

Ben Salem and colleagues studied the in vitro capacity of different extracts (butanolic, ethanolic, and aqueous) of *C. cardunculus* var. *scolymus* L. Fiori to inhibit the activity of *α*-amylase; the results showed that the ethanolic extract exhibited an IC_50_ value of 72.22 μg/mL (compared to acarbose specific inhibitor (IC_50_ = 14.83 μg/mL)). Furthermore, the authors investigated the possible anti-diabetic effect of the ethanol leaf extract of *C. cardunculus* var. *scolymus* L. Fiori in alloxan-induced stress oxidant Wistar rats. The Wistar rats were supplemented with two daily doses of the ALE extract (200 mg/kg or 400 mg/kg) for 28 days. After the supplementation, ALE decreased the *α*-amylase levels in the serum of diabetic rats, which consequently lowered the blood glucose rate. Additionally, the administration of ALE also affected the lipidic profile and antioxidative activity in the liver, kidney, and pancreas of diabetic rats [176]. In addition, the antidiabetic activity of infusions (200 mg/L) of *Agrimonia eupatoria* L. and *C. altilis* L. was evaluated by Kuczmannová et al., by monitoring the inhibitory effect of *α*-glucosidase, serum glucose levels, formation of advanced glycation end-products (AGEs), and the activity of butyrylcholinesterase. The artichoke extract did not affect the activity of *α*-glucosidase, but induced a reduction in the glucose levels and inhibited the formation of AGEs [177]. Similarly, the treatment with an oral administration of ALE (200 and 400 mg/kg) on streptozotocin (STZ)-induced diabetic rats resulted in a significant decrease in serum glucose, TG, and TC levels [178].

In a clinical trial conducted by Ebrahimi-Mameghani and collaborators, the authors examined the effects of supplementation with ALE (1800 mg/daily) in patients with metabolic syndrome for 12 weeks. After the supplementation with ALE, there was a decrease in insulin and in homeostasis model assessment of insulin resistance (HOMA-IR) values [179]. 

These studies confirmed the antidiabetic activity of cardoon, which might be related to the content of chlorogenic acid, which can suppress glucose 6-phosphate, responsible for regulating blood glucose [180,181], and caffeoylquinic acid, responsible for regulating *α*-glucosidase activity [181,182].

**Table 13 foods-11-00336-t013:** Overview of the antidiabetic activity of *Cynara cardunculus*.

Variety/Species	Intervention	Study Type	Model	Result	Ref.
var. *scolymus* L. Fiori	Ethanolic and aqueous extract from outer bracts and the stems	*in vitro*	-	Both ethanolic and aqueous extracts were capable of inhibiting fructosamine formation and antiglycative agents. Moreover, the aqueous extract had a better performance against the systems containing glucose and fructose; on the contrary, the ethanolic extract demonstrated a better activity to inhibit AGE formation when ribose or MGO acted as precursors.	[183]
	Methanolic extract from inedible floral stems	*in vitro* and *in vivo*	Suisse albino mice with alloxan-induced diabetes	The administration of the extract resulted in a decrease in the serum profile levels and blood glucose.	[184]
var. *altilis* DC.	Aqueous extract	*in vitro* and *in vivo*	Wistar rats with STZ-induced diabetes	Both plant extracts presented good anti-glucosidase, antiglycation, and antihyperglycemic properties. In addition, the experiments on isolated aortas exhibited an improvement of vascular dilatory functions in diabetic animals.	[177]
var*. scolymus* L. Fiori	Ethanolic extract from leaves	*in vitro* and *in vivo*	Wistar rats with alloxan-induced diabetes	Both doses of the extract from *C. scolymus* decreased the activity of *α*-amylase, consequently reducing the blood glucose rate.	[176]
	Methanol extract of cereal-based chips enriched with omega-3-rich fish oil and artichoke bracts	*in vitro* and *in vivo*	Suisse Albino mice with alloxan-induced diabetes	In diabetic mice, the enriched chips normalized the levels of blood glucose and serum markers such as alanine aminotransferase, aspartate aminotransferase, alkaline phosphatase, urea, and creatinine. Furthermore, there was an improvement of the serum lipid profile.	[185]
	Butanolic extract from leaves	*in vivo*	Albino rats with streptozotocin (STZ)-induced hyperglycemia	The treatment caused a decrease in the blood sugar levels in the diabetes induced-rats, compared with the STZ diabetic rats.	[186]
	Hydroalcoholic extract from leaves	*in vivo*	Wistar rats with STZ-induced diabetes	In the groups with the administration of artichoke extract, compared with the diabetic group, there was an increase in insulin levels, while the serum concentrations of glucagon and glucose were reduced.	[187]
var. *scolymus* L. Fiori	Ethanolic extract	*in vivo*	Wistar rats with STZ-induced diabetes	When compared with glibenclamide, the use of artichoke extract decreased the lipid profile, increased HLD-c, and decreased HbA1C. However, compared with the group treated with glibenclamide, fasting blood glucose levels were elevated.	[188]
var. *altilis* DC.	Aqueous extract from leaves	*in vivo*	Sprague–Dawley albino rats with induced hypercholesterolemia	The hypercholesterolemia-induced rats treated with both doses of ALE exhibited a decrease in fasting blood glucose, creatinine, uric acid, and urea. These results show that artichoke extracts can be used as a complementary treatment for renal damage and diabetes.	[189]
var*. scolymus* L. Fiori	Hydroalcoholic extract from flowering head	*in vivo*	Wistar rats and obese Zucker rats	*Cynara scolymus* flowering head extract had the capacity to lower the glycemia in both rat strains; however, the extract had a higher efficacy in Wistar rats than in Zucker rats.	[190]
	Luteolin-enriched artichoke leaves	*in vivo*	C57BL/6N mice with high-fatdiet-induced obesity	The treatment with ALE and luteolin-enriched ALE reduced the levels of insulin and glucose.	[150]
	Hydroalcoholic extract from flowering head	*in vivo*	Wistar rats	The combination of both plants extract resulted in a reduction in the food intake, mainly due to the *P. vulgaris* extract, and a decrease in the glycemia.	[181]
var*. altilis* DC.	Hydroalcoholic extract from leaves	*in vivo*	Sprague–Dawley rats with nonalcoholic fatty liver disease induced by high-fat diet	The administration of the extract in a high-fat diet reduced the levels of serum glucose, serum lipid profile, and MDA. Moreover, the 20 mg/kg dose was more effective in completely and significantly restoring OCTN1 and OCTN2 expression in rats fed a high-fat diet.	[160]
var*. scolymus* L. Fiori	Hydroalcoholic extract from leaves	Clinical trial	Patients with metabolic syndrome	The administration of the *Cynara* extract resulted in a decrease in insulin and in HOMA-IR values in patients with the TT genotype of TCF7L2-rs7903146 polymorphism. However, this supplementation did not have any effect toward the blood glucose levels.	[179]

-, there is no data

## 5. Conclusions

The current review provided the nutritional and phytochemical composition of cardoon and its therapeutic applications in metabolic disorders; specifically, hepatoprotective, hypolipidemic, and antidiabetic properties were highlighted. Cardoon is a plant used in the Mediterranean diet and traditional medicine, composed of several bioactive compounds, such as dietary fibers, sesquiterpenes lactones, and phenolic compounds. *C. cardunculus* L. is one of the most promising Mediterranean plants, due to its therapeutic potential and industrial applications. Several studies have demonstrated that cardoon has the capacity to act as an anti-inflammatory, antidiabetic, lipid-lowering, antimicrobial, and antitumoral agent due to its phytochemical composition. It is important to exploit cardoon’s capacity to be applied as an adjuvant in therapies to prevent or diminish the occurrence of diabetes, as well as cholesterol-related diseases. This yields new insights into the advantages of employing other processing and formulation technologies, such as nanoparticles, microencapsulation, and nanoemulsions, to improve their functionality and to increase the use of nutraceutical products, food supplements, and pharmaceutical formulations. Nevertheless, further studies are needed to fully comprehend the mechanism of action of cardoon metabolites underlying the biological activities stated in this review. It is imperative to conduct more clinical trials to clarify the dose and duration of the therapeutics.

## Figures and Tables

**Table 10 foods-11-00336-t010:** Phenolic compounds found in vegetal parts of cardoon.

Phenolic Compounds	Vegetal Parts	References
1-*O*-Caffeoylquinic acid	Leaves (n.d.–0.3 mg/g d.w.) Leaves (2.95–9.53 µm/g d.w.) Stalks (0.6–1.1 mg/g d.w.) Receptacle and bracts (0.3 mg/g d.w.) Florets (n.d.)	[102,114] [113] [114] [114] [114]
3-*O*-Caffeoylquinic acid	Leaves (n.d.–tr mg/g d.w.) Stalks (0.2–0.9 mg/g d.w.) Receptacle and bracts (0.5 mg/g d.w.) Florets (n.d.)	[102,114] [114] [114] [114]
*cis*-5-*O*-Caffeoylquinic acid	Seeds (0.96–35.8 mg/g d.w.) Bracts (n.d.–7.04 mg/g d.w.) Heads (n.d.–3.55 mg/g d.w.) Raw (2.93 mg/g d.w.) Stalks (2.30–3.15 mg/g d.w.)	[46,47] [43] [7] [117] [117]
5-*O*-Caffeoylquinic acid	Stalks (2.30–3.15 mg/g d.w.) Raw (2.93 mg/g d.w.) Leaves (n.d.–2.3 mg/g d.w.) Leaves (18.82–73.68 µm/g d.w.) Leaves extracts (40–63 mg/g d.w.) Leaves extracts (51.3–632 mg/L) Plant (0.743 mg/g d.w.) Heads (n.d.–3.55 mg/g d.w.) Stalks (15.3–17.6 mg/g d.w.) Receptacle and bracts (20.6 mg/g d.w.) Florets (0.9 mg/g d.w.) Seeds (n.d.–2.7 mg/g d.w.)	[117] [117] [102] [113] [118] [13,109] [13] [31] [114] [114] [114] [46,47]
*cis-*1,3-*O*-Dicaffeyolquinic acid	Seeds (n.d.–0.777 mg/g d.w.)	[46,47]
1,3-*O*-Dicaffeyolquinic acid (cynarine)	Stalks (tr mg/g d.w.) Raw (tr mg/g d.w.) Leaves extract (6.5–22.4 mg/g d.w.) Plant (0.0117 mg/g d.w.) Stalks (1.2–1.6 mg/g d.w.) Receptacle and bracts (1.0 mg/g d.w.) Florets (n.d.) Seeds (n.d.–0.68 mg/g d.w.)	[117] [117] [13,118] [13] [114] [114] [114] [46,47]
3,4-*O*-Dicaffeoylquinic acid	Seeds (0.507–6.2 mg/g d.w.) Stalks (tr) Raw (nd) Leaves extract (0.03–2.1 mg/g d.w.)	[46,47] [117] [117] [13,118]
3,5-*O*-Dicaffeoylquinic acid	Seeds (14.8–418 mg/g d.w.) Bracts (0.119–21.83 mg/g d.w.) Heads (0.407–9.9 mg/g d.w.) Stalks (tr–0.1.09 mg/g d.w.) Raw (1.56 mg/g d.w.) Leaves (n.d.–5.75 mg/g d.w.) Plant (0.0575 mg/g d.w.)	[46,47] [43] [7,31] [117] [117] [13,102] [13]
1,4-Dicaffeoylquinic acid	Stalks (tr–0.68 mg/g d.w.) Raw (tr) Stalks (0.9–1.5 mg/g d.w.) Receptacle and bracts (2.7 mg/g d.w.) Florets (n.d.) Leaves (n.d.)	[117] [117] [114] [114] [114] [114]
1,5-Dicaffeoylquinic acid	Stalks (tr–1.09 mg/g d.w.) Raw (1.09 mg/g d.w.) Leaves (n.d.–0.1 mg/g d.w.) Leaves extracts (119.3–230.5 mg/L) Plant (0.827 mg/g d.w.) Stalks (14.3–18.8 mg/g d.w.) Receptacle and bracts (24.5 mg/g d.w.) Florets (4.8 mg/g d.w.)	[117] [117] [13,102,114] [109] [13] [114] [114] [114]
4,5-Dicaffeoylquinic acid	Stalks (n.d.–tr mg/g d.w.) Raw (n.d.) Leaves extract (1.1–5.1 mg/g d.w.) Plant (0.896)	[117] [117] [13,118] [13]
4-Acyl-di-*O*-caffeoylquinic acid isomer	Leaves (n.d.) Stalks (0.7 mg/g d.w.) Receptacle and bracts (0.6 mg/g d.w.) Florets (n.d.)	[114] [114] [114] [114]
Tri-*O*-Caffeoylquinic acid	Heads (n.d.–1.29 mg/g d.w.) Heads (n.d.–1.29 mg/g d.w.)	[7] [31]
Dicaffeoylquinic acids	Leaves (9.44–51.15 µmol/g d.w.)	[113]
*p*-Coumaric acid	Bracts (n.d.–1.4 mg/g d.w.)	[43]
*p*-Coumaric acid hexoside	Heads (n.d.–3.55 mg/g d.w.) Heads (n.d.–1.40 mg/g d.w.)	[7] [31]
Succinyl-diCQA I	Stalks (n.d.–tr) Raw (n.d.)	[117] [117]
Succinyl-diCQA II	Stalks (n.d.) Raw (n.d.)	[117] [117]
Succinyl-diCQA III	Stalks (n.d.) Raw (tr)	[117] [117]
Succinyl-dicaffeoylquinic acid	Leaves (n.d.–8.67 µmol/g d.w.)	[113]
1,5-Di-*O*-caffeoylsuccinoylquinic acid isomer	Leaves (n.d.) Stalks (10.7–12.4 mg/g d.w.) Receptacle and bracts (n.d.) Florets (n.d.)	[114] [114] [114] [114]
4-Acyl-di-*O*-caffeoylsuccinoylquinic acid isomer	Leaves (n.d.) Stalks (2.7–2.8 mg/g d.w.) Receptacle and bracts (n.d.) Florets (n.d.)	[114] [114] [114] [114]
Dicaffeoylsuccinoylquinic acid isomer	Leaves (n.d.) Stalks (1.1–1.5 mg/g d.w.) Receptacle and bracts (n.d.) Florets (n.d.)	[114] [114] [114] [114]
Dicaffeoyldisuccinoylquinic acid isomer	Leaves (n.d.) Stalks (1.6–1.9 mg/g d.w.) Receptacle and bracts (n.d.) Florets (n.d.)	[114] [114] [114] [114]
Eridictyol-*O*-glucuronide	Bracts (n.d.–0.98 mg/g d.w.)	[43]
Eriodictyol hexoside	Leaves (n.d.) Stalks (n.d.) Receptacle and bracts (n.d.) Florets (0.1 mg/g d.w.)	[114] [114] [114] [114]
*p-*Coumaroylquinic acid	Leaves extracts (0.09–1.1 mg/g d.w.)	[118]
5-*O*-Feruloylquinic acid	Bracts (n.d.–0.55 mg/g d.w.) Leaves extracts (0.6–1.6 mg/g d.w.)	[43] [118]
Scopolin isomer	Leaves (n.d.) Stalks (n.d.) Receptacle and bracts (n.d.) Florets (1.2 mg/g d.w.)	[114] [114] [114] [114]
Kaempferol-3-*O*-rutinoside	Bracts (n.d.–0.59 mg/g d.w.)	[43]
Luteolin-*O*-glucuronide	Bracts (n.d.–2.036 mg/g d.w.) Heads (n.d.–1.03 mg/g d.w.) Leaves (n.d.–2.4 mg/g d.w.) Leaves (n.d.–0.48 µmol/g d.w.) Stalks (0.4–1.0 mg/g d.w.) Receptacle and bracts (0.8 mg/g d.w.) Florets (0.6 mg/g d.w.)	[43] [7,31] [102] [113] [114] [114] [114]
Luteolin-7-*O*-glucuronide	Heads (n.d.–0.90 mg/g d.w.) Leaves (n.d.–13.7µmol/g d.w.) Leaves extracts (10.9–189.4 mg/L)	[7,31] [113] [109]
Luteolin-*O*-hexoside	Bracts (n.d.–1.51 mg/g d.w.)	[43]
Luteolin acetyl-hexoside	Leaves (0.7 mg/g d.w.) Stalks (n.d.) Receptacle and bracts (n.d.) Florets (0.3 mg/g d.w.)	[114] [114] [114] [114]
Luteolin-7-*O*-rutinoside	Leaves (n.d.–6.5 mg/g d.w.) Leaves (n.d.–0.48 µmol/g d.w.) Stalks (n.d.) Receptacle and bracts (n.d.) Florets (0.3 mg/g d.w.)	[102,114] [113] [114] [114] [114]
Luteolin glucoside	Leaves (n.d.–4.2 mg/g d.w.) Leaves extracts (7.4–9.3 mg/g d.w.)	[102] [118]
Luteolin-7-*O*-glucoside (cynaroside)	Leaves (0.66–33.55 µmol/g d.w.) Leaves extract (2.9–3.8 mg/g d.w.) Stalks (0.3–1.2 mg/g d.w.) Receptacle and bracts (0.4 mg/g d.w.) Florets (0.6 mg/g d.w.)	[113] [114,118] [114] [114] [114]
Luteolin-*O*-malonyl-hexoside	Bracts (n.d.–1.265 mg/g d.w.) Leaves (n.d.–2.2 mg/g d.w.)	[43] [102]
Luteolin-7-*O*-malonyl-hexoside	Leaves (1.11–43.00 µmol/g d.w.) Leaves extracts (1.0–1.7 mg/g d.w.) Leaves extracts (n.d.–83.3 mg/L)) Heads (n.d.–1.17 mg/g d.w.)	[113] [118] [109] [31]
Luteolin	Leaves (0.02–2.02 µmol/g d.w.) Stalks (n.d.) Receptacle and bracts (n.d.) Florets (1.2 mg/g d.w.)	[113,114] [114] [114] [114]
Apigenin-7-*O*-glucuronide	Bracts (2.79–10.6 mg/g d.w.) Heads (n.d.–13.2 mg/g d.w.) Leaves (n.d.–2.4 mg/g d.w.) Leaves (n.d.–2.31 µmol/g d.w.) Leaves extracts (n.d.–115.0 mg/L)	[43] [7,31] [102] [113] [109]
Apigenin glucuronide	Leaves (2.9 mg/g d.w.) Stalks (1.2–1.4 mg/g d.w.) Receptacle and bracts (8.1 mg/g d.w.) Florets (13.8 mg/g d.w.)	[114] [114] [114] [114]
Apigenin-7-*O*-glucuronide-hexoside	Heads (n.d.–1.07 mg/g d.w.)	[7,31]
Apigenin	Leaves (n.d.–6.79 µmol/g d.w.) Leaves extracts (n.d.–3.8 mg/L) Stalks (n.d.) Receptacle and bracts (n.d.) Florets (4.8 mg/g d.w.)	[113] [109,114] [114] [114] [114]
Apigenin-*O*-malonyl-hexoside	Bracts (n.d.–1.77 mg/g d.w.)	[43]
Apigenin acetyl-hexoside	Leaves (0.5 mg/g d.w.) Stalks (0.1–0.3 mg/g d.w.) Receptacle and bracts (0.6 mg/g d.w.) Florets (0.7 mg/g d.w.)	[114] [114] [114] [114]
Apigenin-7-*O*-rutinoside	Heads (0.99–3.51 mg/g d.w.) Leaves (n.d.–0.7 mg/g d.w.) Leaves (n.d.–1.39 µmol/g d.w.)	[7,31] [102] [102]
Apigenin-7-*O*-glucoside	Leaves (n.d.–0.1 mg/g d.w.) Leaves extracts (n.d.–61.7 mg/L)	[102] [109]
Apigenin-malonylhexoside	Heads (n.d.–1.17 mg/g d.w.) Leaves extracts (n.d.–72.1 mg/L)	[7] [109]
Naringenin-7-*O*-glucoside	Leaves (n.d.) Stalks (n.d.) Receptacle and bracts (n.d.) Florets (3.2 mg/g d.w.)	[114] [114] [114] [114]
Naringenin rutinoside	Leaves (n.d.) Stalks (n.d.) Receptacle and bracts (n.d.) Florets (5.4 mg/g d.w.)	[114] [114] [114] [114]
Naringenin	Leaves (n.d.) Stalks (n.d.) Receptacle and bracts (n.d.) Florets (0.2 mg/g d.w.)	[114] [114] [114] [114]
Chrysoeriol hexoside isomer	Leaves (0.5 mg/g d.w.) Stalks (n.d.) Receptacle and bracts (n.d.) Florets (0.2 mg/g d.w.)	[114] [114] [114] [114]
Chrysoeriol isomer	Leaves (n.d.) Stalks (n.d.) Receptacle and bracts (n.d.) Florets (0.1 mg/g d.w.)	[114] [114] [114] [114]

tr, traces (below the limit of quantification); n.d., not detected.

## Data Availability

Data are contained within this article.

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
