# Peer review of "Bioactive Compounds from Cardoon as Health Promoters in Metabolic Disorders"

_foods, 2022, doi:10.3390/foods11030336_

Round 1
Reviewer 1 Report
The authors can find comments and suggestions (highlighted) in the attached file.
Also, I want to suggest modifying table 9
Table 9. Volatile organic compounds found in vegetal parts of cardoon.
|
Volatile organic Vegetal parts References |
||
|
Aromatic compound |
||
|
Benzoic acid |
|
|
|
2,6-Dimethoxyhydroquinone |
|
|
|
Vanillylpropanoic acid |
|
|
|
|
Stalks, receptacle and bracts, florets |
[42] |
|
|
|
|
|
Vanillin |
|
|
|
3-Vanillylpropanol |
Leaves |
[42] |
|
1-Octen-3-one |
|
|
|
6-Methyl-5-hepten-2-one |
|
|
|
Octanal Benzene acetaldehyde |
|
|
|
|
Leaves |
[103] |
|
|
|
|
|
|
|
|

Author Response
The authors would like to thank the reviewer for all the important suggestions and comments. The manuscript was revised considering all the comments and suggestions made by the reviewer. Furthermore, table 9 was modified to facilitate its reading.

Reviewer 2 Report
The authors prepared an extensive overview on the topic »Cardoon dietary bioactive compounds as health promoters in metabolic disorders. « The cardoon nutritional and phytochemical composition and biological potential are provided. The main therapeutical effects of the different parts of the plant on metabolic disorders, namely associated with hepatoprotective, hypolipidemic, and antidiabetic activity, were exposed. The authors gave an extensive and appropriate introduction. In review, state of the art regarding cardoon nutritional and phytochemical composition and biological potential is shown and discussed. The most recent literature was very well researched and discussed.
The current review is an essential contribution in this area. The manuscript is well written and organized. The number of references is sufficient. There are a lot of tables with very interesting information. An overview of the antidiabetic, hypolipidemic, hepatoprotective activity of Cynara Cardunculus with results is given in the tables. Tables are also well presented and understandable. The text is understandable and readable.
The topic is interesting because it gives a better understanding of cardoon dietary bioactive compounds as health promoters in metabolic disorders. The current review offered cardoon's nutritional and phytochemical composition and its therapeutic applications in metabolic disorders, such as hepatoprotective, hypolipidemic, and anti-diabetic properties were exposed. With review authors well explained that cardoon has the capacity to act as an anti-inflammatory, antidiabetic, lipid-lowering, antimicrobial, and anti-tumoral agent due to their phytochemical composition. These insights also open new insights towards the advantages of utilizing other processing and formulation technologies.
In my opinion, the review article is very good and needs an English check by a native speaker.
Author Response
The authors thank the reviewer for the comments. The manuscript was carefully revised. Please see the revised version of the manuscript.
Reviewer 3 Report
- The current work aims to provides the state of the art regarding cardoon nutritional and phytochemical composition in addition to its biological activities
- The followings are some comments and recommendations:
- I recommend the following title:
Bioactive compounds from Cardoon as health promoters in metabolic disorders
- In Key-words: plz replace biological activities instead of biological activity
- I ask author if obtained the permission for the used data and results such as Table 1 and table 8
- I ask author for checking again the percent of primary metabolites content in table 1 (mentioned t. protein is 0.7%, t. lipids is 0.1 and t. carbohydrates is 4%) very little . so, plz check again.
- I need more clarification for total carbohydrate content in Table 1 and total fiber in Table 3. (plz check and clarify).
- The authors mention most of bioactive compounds in cardoon plant with its nutrition values except amino acids types and content (very important in nutrition), so, I ask authors to provide the amino acids composition of plant under study.
- Plz check again the total percentage of fatty acids content in Table no. 4 (the sum not equal to 100%).
- Plz provide any information’s related to cardoon toxicity or its side effects on living cells
Author Response
The authors would like to acknowledge the reviewers for the valuable comments that contributed to further improve the impact and value of this manuscript. Furthermore, the manuscript was carefully revised, and all the recommended modifications were performed (marked using “track changes” in the main manuscript). Therefore, the authors trust that the manuscript is now suitable for publication in the Journal Foods.
The current work aims to provides the state of the art regarding cardoon nutritional and phytochemical composition in addition to its biological activities
The followings are some comments and recommendations:
- I recommend the following title:
Bioactive compounds from Cardoon as health promoters in metabolic disorders
The title was changed as requested. Please see the revised version of the manuscript.
- In Key-words: plz replace biological activities instead of biological activity
The authors thank the reviewer for the suggestion. The suggestion was performed and marked in the manuscript (please see page 1, line 27).
- I ask author if obtained the permission for the used data and results such as Table 1 and table 8
The authors acknowledge the reviewer’s comment, the data present in both tables is from the USDA website. The information present on this website is public and since we only use the data, it is only necessary to cite the source.
- I ask author for checking again the percent of primary metabolites content in table 1 (mentioned t. protein is 0.7%, t. lipids is 0.1 and t. carbohydrates is 4%) very little . so, plz check again.
The authors thank the reviewer for the observation. The data presented in table 1 is in agreement with the source. Please see the website of USDA:http://fdc.nal.usda.gov/foodcomposition.
- I need more clarification for total carbohydrate content in Table 1 and total fiber in Table 3. (plz check and clarify).
The values of total carbohydrate are in accordance with the original source. Please see the website of USDA:http://fdc.nal.usda.gov/foodcomposition.Cardoon presented 94% of water, and 6% of dry residue (composed by sugars, protein, ash, fattty acids, etc.).
In respect to the total fiber of the dy residue, cardoon chemical composition revealed high content of polysaccharides (more than 50%) being the major cellulose (41.9). The fiber composition of raw, stems, stalks, leaves and flowers are presented in Table 3. Remember that, their contents are relative to the dry residue (that represent about 6% of fresh cardoon – Table 1).
Please see the cited references:
Gominho, J.; Curt, M.D.; Lourenço, A.; Fernández, J.; Pereira, H. Cynara cardunculus L. as a biomass and multi-purpose crop: A review of 30 years of research. Biomass and Bioenergy 2018, 109, 257-275, doi:https://doi.org/10.1016/j.biombioe.2018.01.001.
Francaviglia, R.; Bruno, A.; Falcucci, M.; Farina, R.; Renzi, G.; Russo, D.E.; Sepe, L.; Neri, U. Yields and quality of Cynara cardunculus L. wild and cultivated cardoon genotypes. A case study from a marginal land in Central Italy. European Journal of Agronomy 2016, 72, 10-19, doi:https://doi.org/10.1016/j.eja.2015.09.014.
Ben Amira, A.; Blecker, C.; Richel, A.; Arias, A.A.; Fickers, P.; Francis, F.; Besbes, S.; Attia, H. Influence of the ripening stage and the lyophilization of wild cardoon flowers on their chemical composition, enzymatic activities of extracts and technological properties of cheese curds. Food Chem 2018, 245, 919-925, doi:10.1016/j.foodchem.2017.11.082.
Fernandes, M.C.; Ferro, M.D.; Paulino, A.F.C.; Mendes, J.A.S.; Gravitis, J.; Evtuguin, D.V.; Xavier, A.M.R.B. Enzymatic saccharification and bioethanol production from Cynara cardunculus pretreated by steam explosion. Bioresource Technology 2015, 186, 309-315, doi:https://doi.org/10.1016/j.biortech.2015.03.037.
Shatalov, A.A.; Morais, A.R.C.; Duarte, L.C.; Carvalheiro, F. Selective single-stage xylan-to-xylose hydrolysis and its effect on enzymatic digestibility of energy crops giant reed and cardoon for bioethanol production. Industrial Crops and Products 2017, 95, 104-112, doi:https://doi.org/10.1016/j.indcrop.2016.10.017.
Giannoni, T.; Gelosia, M.; Bertini, A.; Fabbrizi, G.; Nicolini, A.; Coccia, V.; Iodice, P.; Cavalaglio, G. Fractionation of Cynara cardunculus L. by Acidified Organosolv Treatment for the Extraction of Highly Digestible Cellulose and Technical Lignin. Sustainability 2021, 13, 8714.
Bertini, A.; Gelosia, M.; Cavalaglio, G.; Barbanera, M.; Giannoni, T.; Tasselli, G.; Nicolini, A.; Cotana, F. Production of Carbohydrates from Cardoon Pre-Treated by Acid-Catalyzed Steam Explosion and Enzymatic Hydrolysis. Energies 2019, 12, 4288.
Shatalov, A.A.; Pereira, H. Dissolving grade eco-clean cellulose pulps by integrated fractionation of cardoon (Cynara cardunculus L.) stalk biomass. Chemical Engineering Research and Design 2014, 92, 2640-2648, doi:https://doi.org/10.1016/j.cherd.2014.01.007.
- The authors mention most of bioactive compounds in cardoon plant with its nutrition values except amino acids types and content (very important in nutrition), so, I ask authors to provide the amino acids composition of plant under study.
The authors thank the reviewer for raising this important observation. However, there is no detailed information in the literature about the amino acids’ composition of C. cardunculus. In the future, these assays could be performed.
- Plz check again the total percentage of fatty acids content in Table no. 4 (the sum not equal to 100%).
The total represents the total variations in all samples reported by the different authors. to facilitate understanding, we have included in the table total variation. Please find the Table 3 of the revised version.
- Plz provide any information’s related to cardoon toxicity or its side effects on living cells
The authors thank the reviewer for raising this important observation. The information was included in the manuscript (please see page 3, from line 122 to line 125).
Reviewer 4 Report
Dear Authors,
I have read your manuscript entitled "Cardoon dietary bioactive compounds as health promoters in metabolic disorders," which describes the nutritional and phytochemical composition and therapeutic value of Cardoon. Major revisions are needed. In my opinion, too much basic or general knowledge is provided. Also, I suggest that you add a paragraph in the conclusion about your critical evaluation, something about what is the state of research on C. cardunculus, what knowledge gaps were found. The specific comments and suggestions can be found in the manuscript file. In addition, the manuscript contains grammatical and syntactical errors that need to be corrected.

Author Response
The authors thank the reviewer for all the relevant comments. All the comments and suggestions were carefully revised and marked in the manuscript. Additionally, a paragraph was added to the conclusion (please see page 34, lines 703 to 705). Furthermore, the manuscript was also carefully reviewed to correct any typo, grammatical and syntactical error.

Round 2
Reviewer 4 Report
The authors of the manuscript entitled 'Cardoon dietary bioactive compounds as health promoters in metabolic disorders' have adequately addressed the points raised. However, I suggest to the authors to once again carefully check the English language throughout the text.